# SANE (Easy Gait Analysis System): Towards an AI-Assisted Automatic Gait-Analysis

**DOI:** 10.3390/ijerph191610032

**Published:** 2022-08-14

**Authors:** Dario Sipari, Betsy D. M. Chaparro-Rico, Daniele Cafolla

**Affiliations:** 1Department of Control and Computer Engineering, Mechatronic Engineering, Politecnico di Torino, 10129 Torino, Italy; 2Biomechatronics Lab, IRCCS Neuromed, 86077 Pozzilli, Italy

**Keywords:** human biomechanics, automated gait analysis, artificial intelligence, motion tracking, markerless

## Abstract

The gait cycle of humans may be influenced by a range of variables, including neurological, orthopedic, and pathological conditions. Thus, gait analysis has a broad variety of applications, including the diagnosis of neurological disorders, the study of disease development, the assessment of the efficacy of a treatment, postural correction, and the evaluation and enhancement of sport performances. While the introduction of new technologies has resulted in substantial advancements, these systems continue to struggle to achieve a right balance between cost, analytical accuracy, speed, and convenience. The target is to provide low-cost support to those with motor impairments in order to improve their quality of life. The article provides a novel automated approach for motion characterization that makes use of artificial intelligence to perform real-time analysis, complete automation, and non-invasive, markerless analysis. This automated procedure enables rapid diagnosis and prevents human mistakes. The gait metrics obtained by the two motion tracking systems were compared to show the effectiveness of the proposed methodology.

## 1. Introduction

Artificial intelligence (AI) is a subfield of computer science that explores the theoretical basis, approaches, and strategies for developing hardware and software systems capable of providing computers with skills that seem to be exclusive to human intelligence [1]. Researchers in artificial intelligence seek to develop computer systems capable of performing intellectual tasks while also aiming to explain underlying conceptions of intelligence. Rather than replicating or simulating human intelligence, the discipline attempts to recreate or mimic it. AI reveals itself to be a fast-growing field, both as a whole and in its constituent parts. There is a constant pragmatic enlargement of the discipline’s bounds whenever new findings are obtained. The variety of approaches is what has remained of early AI’s characterization to this day.

This study focuses on machine learning and one of its subsets: deep learning. Deep learning makes use of the branching of neural networks to generate a network with numerous layers of processing [2].

Numerous studies have been published in the literature in recent years investigating the application possibilities of artificial intelligence [3,4,5,6,7,8,9,10,11,12,13,14,15,16,17]. Medicine plays a significant role in this expansion. AI-based technologies have the potential to significantly increase the use of many diagnostic techniques in primary care. The digitalization of health data, in conjunction with computer analytics, provides intelligent and efficient chronic patient treatment [7]. Additionally, it expands the scope of clinical research, frontier medicine, and customized medicine. 

Deep learning methods were shown to be superior in terms of sensitivity and specificity for ECG-based diagnosis of acute coronary syndrome [8]. It has even been established that cardiology medical personnel have a statistically higher diagnosis accuracy in the case of arrhythmia patterns in electrocardiographic tracings [9]. AIs are capable of “reasoning” about data in ways that human senses cannot. Emotion recognition in conversation (ERC) is gaining traction as a new field of study in natural language processing (NLP). The capacity to notice and link patterns shows promise for partly replicating emotional intelligence. Additionally, ERC is critical for creating emotion-aware conversations that need a grasp of the user’s emotions [14].

In this context, medical doctors and therapists employ artificial intelligence (AI) in motion capture (MoCap) to aid diagnosis and treatment of their patients. Motion capture is the process of digitally capturing and mapping human motions into a three-dimensional digital model. One of the most promising approaches makes use of artificial intelligence to distinguish an individual’s joints.

### 1.1. AI for Pose Estimation

The process of acquiring information on an individual’s joints in their entirety is referred to as “skeleton tracking.” This is because, in an ideal world, a virtual skeleton is projected onto the collected body picture and the three-dimensional position of each joint is monitored in order to recreate the subject’s spatial mobility. The most current approach for this sort of collection takes use of artificial intelligence technologies such as Cubemos, Nuitrack, and OpenPose:
PoseNet is an artificial intelligence programmed in Python using TensorFlow, a widely known machine learning framework created and maintained by Google. PoseNet recognizes 17 essential points: the eyes, ears, nose, shoulders, knees, hips, elbows, and ankles. PoseNet is capable of running on both CPUs and GPUs. This tool is available in two modes: a single-person pose detector that is quicker and easier to use but needs just one subject in the picture, and a multiple-person pose detector. One of the most compelling features of this skeleton tracking tool is its capacity to function nearly independently of the number of subjects evaluated concurrently [18].OpenPose is a free and open-source project. Currently, the framework supports two models: the COCO MPI model (which is quicker but less accurate) and the BODY 25 model. This skeleton tracking program is one of the few that can offer a greater and more comprehensive number of points, including critical spots on the feet and their angle [19].Nuitrack is a private closed-source application. It is a deep learning-based artificial intelligence. It is cross-platform, having capabilities for 3D full-body skeleton tracking, gesture recognition, and face tracking. Nuitrack detects 19 distinct points: it does not track the feet but does track the hands [20].Cubemos’ Skeleton Tracking SDK is a closed-source proprietary application that utilizes Intel’s Distribution of OpenVINO toolset. It is a machine learning-based artificial intelligence. It is a real-time, cross-platform, multi-person, 3D full-body posture estimator, although its use should be limited to scenes including up to five individuals. Cubemos recognizes the following 18 key points: eyes, ears, nose, shoulders, knees, wrists, hips, elbows, ankles, and spine. As a result, it is unable to offer information on the hands and feet or certain rotations. It is capable of using both CPUs and GPUs. Cubemos’ Skeleton Tracking SDK is optimized for use with Intel devices as an Intel partner [21].

### 1.2. General-Purpose Technologies for Motion Acquisition

MoCap finds application in the study topic of this publication: gait analysis.

Gait analysis has a variety of uses in the neurological realm as well. The gait cycle (GC) is the basic unit of gait. It is defined as the time interval between a foot’s first contact with the ground and its subsequent contact with the surface.

Kinematic and kinetic parameters, along with ground reaction forces, comprise the dataset that completes the gait analysis. This set gives a clinical assessment for gait cycle analysis. Estimates of these parameters based on kinematics and anthropometric data (including weight) are sometimes utilized but lack the precision and accuracy of direct measurement.

Numerous methodologies may be used to measure human movement and its variations. Qualitative or quantitative evaluations may be conducted using a variety of different systems or tools.

In recent years, gait analysis has transitioned from a qualitative approach toward one that is nearly totally quantitative.

Two broad categories of quantitative analysis tools exist: optical and non-optical systems [19,22].
Non-optical:
Instruments for magnetic, inertial, or electromechanical kinematic analysis. Electro-goniometers that detect 2D and 3D angular joint movement; accelerometers that compute up to six degrees of freedom of angular and linear joint movement.Pressure and dynamometric plates for dynamic analysis. Strain gauges or piezoelectric transducers are used to obtain information on the stresses and motions of the route [23].Electromyographic sensors for electromyographic analysis (EMG). They evaluate the action potentials of motor units and the neuromuscular activity responsible for movement [23].Optical:

Optoelectronic systems based on stereophotogrammetry. At the HW level, pictures are captured via cameras that can capture images at multiple light frequencies. Here, the number of cameras used is dependent upon the analysis and ranges between 10 and 50.

At least two devices are required for 3D reconstruction utilizing picture triangulation or information from several spectra (such as a depth camera plus RGB). The greater the number of instruments used, the higher the frequency and quality of acquisitions without the need for complex reconstructions or software filtering.

By means of these systems, it is possible to monitor the patient’s three-dimensional movements by tracking the placement of the markers over time.

Markers can be passive or active:Passive markers are typically spherical or hemispherical in shape and coated with a retroreflective material to reflect light or a certain frequency.Active markers, on the other hand, are powered by a battery and incorporate an LED (generally producing infrared light, halving the distance it has to travel to be captured by cameras). However, they are double the size and expense of the passive alternative.

These instruments are widely described and categorized in the overview of Muro-De-La-Herran [23] from which Table 1 and Table 2 are extracted.

As is natural to deduce, each gait analysis device has distinct advantages and limitations. Non-optical devices are prone to operator placement errors since they must be placed directly on the human body. In addition, tissue movement causes sampling problematic because of friction. No magnetic, inertial, or electromechanical equipment is compatible with all patients or examination areas. Many of them, particularly electro-goniometers, are susceptible to crosstalk. While dynamometric platforms are not susceptible to wearability difficulties, they are constrained by the sample perimeter: in order to produce a heel strike, the patient must perfectly center the acquisition plates with their foot. The measures are inexpensive and do not need a large area.

Normally sEMG sensors are employed in combination with other sensors to detect muscle activation while walking. sEMG sensors, due to their highly disturbed signal, can be sometimes invasive, and sometimes they require special treatment, such as skin demarcation during anthropometric measurements.

Dynamometric platforms, particularly the most advanced ones, considerably help the detection of gait phases by generating dynamic data and vector fields in the stance phase, typically with high accuracy.

These are the reasons why current research focuses on identifying less noticeable and more ergonomic approaches that do not alter measurements and combine analysis with normal activities to minimize the stress of the patient and expand the dataset on movements performed under a variety of conditions.

It is still possible to estimate several non-optical data using the accelerometers and gyroscopes in mobile phones, allowing a preliminary study of movement performance which is not clinically adequate but useful for an initial investigation.

As accuracy and complexity increase, it becomes possible to use truly portable and wearable technology in everyday life.

These non-optical systems belong to a separate category: they are more complex, contemporary, and advanced, and they are often connected with software at several levels, ranging from low-level hardware management to high-level data processing and reconstruction, which rarely use machine learning techniques for this purpose.

Some examples are: Xsens MVN inertial sensors or the wireless M3D gait analysis system developed by Tec Gihan Co [29], flexible platforms equipped with sensors to overcome the limitations of dynamometer plate frames, or soles equipped with sensors such as piezoresistive pressure sensors by FlexiForce [23] or Veristride insoles to overcome both criticalities of fixed dynamometer plates.

However, per capita costs and consumption for a gait analysis are raised; the signals are always vulnerable to external disturbances and motion; and the power supply, size, and battery life continue to be limiting factors. Moreover, real-time data are provided at a slower sample rate as with the UEFS smart-walker with MARG sensors due to the hardware limitations of a compact wearable device.

The category of systems using quantitative, semi-subjective, or fully qualitative approaches has its own universe, with various advantages and disadvantages that change depending on circumstances. In general, as in the case of the research of Pirlo, G., and Luigi [30], the T25-Fw technique, the MSWS-12 method, and the POMA test [31,32,33,34] are examples of procedures that cannot be considered sufficient for a comprehensive review owing to their lack of comparability and reproducibility. Lastly, their reliability and precision vary depending on the population under research.

There are several kinds of optical systems, including those previously described, that exist in a variety of forms, but the majority share the same advantages and disadvantages. Stereophotogrammetry and optoelectronic systems that rely only on computer vision software are more susceptible to human, instrumental, and systemic setup errors. These systems have many of the same problems as wearable systems due to passive and active markers; however, these weaknesses may be rectified by supporting software and a variety of visual feedback. They are often highly expensive, require specialized equipment and personnel, have long analysis times, and do not permit non-clinical observations.

The acquisition phase of these systems has a high sample rate and a high degree of precision. Infrequently, signal reconstruction is required due to technology that synchronizes data from several cameras.

Many new, high-tech alternatives to the usual methods have arisen in recent years. Automation seeks to evaluate gait while minimizing human error. In some cases, researchers have tried to give data to identify disease-related gait patterns without a full gait study. Academic research on Parkinsonian traits and associated neurodegenerative illnesses is one example. Here offline image processing and the open-source AI TensorFlow were applied. The final product automatically calculates upper body, arm, and knee inclination rates in percentage. These behaviors cannot be associated with standard gait analysis, but they are qualitatively helpful indicators and, above all, a type of automation [30].

### 1.3. Gait-Analysis-Focused Technologies

G.O.A.L.S. is one of the products considered for academic and commercial purposes. This lab’s initiatives reach high levels of automation while maintaining a clinically acknowledged output.

G.O.A.L.S. extends the features of a complete 3D parametric biomechanical skeleton model, developed in an original way for static 3D posture analysis, to kinematic and kinetic analysis of movement, gait, and run. By integrating baropodometric data, the model allows the estimation of lower limb net joint forces, torques, and muscle power. Net forces and torques are also assessed at intervertebral levels [35]. Acquisition requires a preparatory phase that includes anthropometric measurements, placing of the markers on the patient, and setup of the clinical procedure connected with the marker arrangement. In the portable system, a standing position acquisition is employed to rebuild all the markers’ relative 3D positions to minimize information loss in walking blind zones. Meanwhile, the dynamic measurement gives not only the total vector of the standing posture, but also a vector field of the weight distribution on the ground. The equipment utilized is redundant and has a high sample frequency to provide excellent data accuracy and dependability, particularly for the non-portable system [35].

Finally, there are kinds of automation that employ wearable sensors rather than computer vision, such as the project conducted in the Intelligent Automation Laboratory of the Department of Electrical Engineering at the Federal University of Espirito Santo [36].

In the case of more advanced optical systems, such as those proposed by G.O.A.L.S., the benefit of simpler, automated, comparatively quicker, and marginally less expensive investigations is beginning to emerge. The decision to continue using marker and computer vision systems results in not only high levels of output quality and precision, but also critical difficulties related to patient preparation stages and those inherent to markers. Automation in the analysis phase drastically decreases subjective operator mistakes, boosts test repeatability, optimizes acquisitions, and exploits the maximum number of possible gait cycles.

The most well-known gait analysis systems that use gold standard protocols are Vicon, Optotrack, and BTS Motion Analysis Lab.

The BTS GAITLAB, for example, is a laboratory model specializing in motion tracking. It employs a minimum of 8 infrared cameras, 6 sensor platforms, 8 electromyographic sensors, 2 RGB cameras, a minimum of 20 markers, and computer vision software [37,38,39,40,41]. For the acquisitions, the system adheres to well-established clinical guidelines from the literature [42,43]. Each acquisition is preceded by two stages of preparation: one is performed daily for system recalibration, and the other is performed before the analysis of each patient.

The recalibration step is divided into sections dedicated to cameras, pressure plates, and EMGs.

The first sub-phase is divided into a dynamic section that consists of multiple “empty” acquisitions of an operator moving a marker-containing wand and a static section in which a new system of Cartesian axes is manually rearranged through video recordings of an object consisting of three identical wands arranged orthogonally to each other and generally positioned at a standard location in the room to make the analyses compatible.

The second sub-phase consists of informing the processing software of the exact positioning of the pressure plates in reference to the previously created Cartesian plane.

The third and final sub-step is decided by whether EMG sensors are required or not. It involves their activation and a fast check of the proper operation of each emitter and receiver.

Each session’s preliminary phase consists of preparing the specific setup for the patient’s study. It requires the collection of anthropometric measures and the simultaneous selection and demarcation on the patient’s skin of the regions that will be covered by EMG sensors. It is occasionally associated with simultaneous depilation of the involved areas for an optimal acquisition of muscle signals.

This is followed by the phase of marker placement, which must conform to the clinical procedure selected for the examination of the patient’s photos.

This is the most important stage of the setup. The protocol is then indicated in the analysis software (in recent years, new protocols specializing in the joint evaluation of back movements have arisen, although the Davis protocol or a variant of it remains the most used).

The walk is then recorded, and the foot contact event and corresponding optimal gait cycle are manually chosen offline. Each recording utilizes the data from a single step cycle (the most accurate according to the operator).

Finally, it is confirmed that each foot has correctly and completely landed on the area inside the platform’s frame, and the software manually indicates which foot has made contact with the platform.

The analysis session concludes with the development of a file summarizing the average gait cycle behavior of the patient.

In order to analyze the problems with gold standard analytical methodologies, the BTS will be utilized as a typical system below.

The first phase is the calibration phase. The negative characteristics of this stage are among the least significant. Both the reset phase of the Cartesian system and the camera check can require variable time to ensure that the complete acquisition volume perceives all markers properly with the appropriate conventions, as similar as possible to those used for the previous tests. This step may be not performed regularly for convenience, but at the cost of a loss in rigor and an increased risk of incorrect acquisition during analysis phases.

The following phase is the patient preparation phase. Here, extra time is devoted to not only anthropometric measurements and body delineation, but also, if needed, sEMG signal verification. In this specific context, the most crucial problems emerge in the definition of these measurements, which are often made, as a rule of thumb, using insufficient equipment, such as basic rulers. This adds multiple orders of measurement error that are not insignificant, such as instrument sensitivity, parallax, procedural mistakes, and human errors. This is compounded by the growing stress placed on the patient by this thorough procedure, which extends to the long palpation phase, which is often performed while the patient is standing for marker placement. Preparation of the patient requires multiple motions and flexions to properly identify the key points and joints. Validation of the right alignment of the markers is similarly imprecise, seldom accomplished using laser beams projected onto the patient’s skin and much more often with basic rulers.

The analytical step is not exempt from complications. After acquiring the patient’s standing pose for calibration, the dynamic and kinematic gait analysis is conducted. In the dynamic component, the acquisition plates present an additional problem: the patient is forced to walk in an unnatural way, trying to center the load cell frames with their foot, causing measurement falsification, or the heel strike occurs randomly on non-sensitive points on the floor, forcing the operator to discard a step or an entire acquisition and ask for a measurement repetition. Due to the markers, in the kinematic component, problems are identical. Many patients are required to perform unnatural motions in order to prevent excessive displacement of the markers, which may make the analysis seem unnatural. The most troublesome are those in the quadriceps and calf region, which are tied with bands that are sometimes too tight, sometimes too loose, and which often become misaligned due to contact with the inner thigh. Changing the patient’s pattern necessitates a new acquisition even in this case. In addition, real-time feedback is not comprehensive and is discouraged by the BTS after the first measurements in order to prevent adding further stress to the test subject. Finally, one and only one step is chosen from each acquisition (the best according to the subjective preference of the operator), and the identification of gait cycle events is entirely manual and recursive. The analysis time is further lengthened; moreover, the criticalities are not restricted to numerical, economic, or procedural variables, nor do they impact only the convenience sphere; rather, a much more significant problem emerges: many patients, particularly the most vulnerable and those who need gait analysis the most, are physically unable to complete the test, returning home exhausted and without findings. Furthermore, there is a very limited guarantee that, despite several measures, all patient data are eventually complete and available.

In conclusion, with the standard stereophotogrammetry method, users deal with a system designed for extremely precise, reliable, and highly sampled measurements, which paradoxically requires the use of imprecise procedures, tools, and preliminary information. Nevertheless, the path marked out by automation methods paves the way for new forms of innovation and accurate, user-friendly measurement.

For all these reasons, in this paper, a new approach to gait analysis is proposed, improving the already existing SANE system described in [37]. SANE is a complete, non-intrusive, and low-cost system for assessing spatiotemporal gait paraments with great potential for assessing spatiotemporal parameters. The main advantage of SANE is that it automatically detects each gait cycle and automatically calculates spatiotemporal gait parameters. In this paper, the SANE system is improved by introducing AI, and the performance of the system with the new approach is evaluated by comparing it with BTS as the gold standard.

## 2. Materials and Methods

This project’s objective is to develop a low-cost gait analysis tool. This study proposes a unique AI-based automated motion characterization method that incorporates real-time, non-invasive analysis and complete automation. This automated technology offers rapid diagnosis while preventing human error, with the goal of offering an alternate, markerless, portable testing instrument relevant to the needs of the most fragile patients.

### 2.1. Procedure

As an intuitive system, SANE enables the acquisition of scalar and kinematic data with a few simple clicks. Before the actual acquisition, a blank recording is recommended for the sole purpose of aligning the camera (4) with the selected analysis region. Referring to Figure 1, each patient’s motions are monitored and analyzed by SANE with the support of artificial intelligence modules (5).

In particular, the operator in charge of the analysis (3) activates the system through its interface (2), which generates the patient record (9) and activates a stream of information from the RGBD camera (4). Each frame is then reassembled using a two-dimensional projection of the patient’s virtual skeleton’s key points after the RGB data are computed in real time by the AI (5) provided by Cubemos. The data obtained with the depth camera (4) are combined with the information in (5) to deduce the third dimension of each point. This whole process is shown as a video to the operator, who can evaluate the quality of the acquisition using a real-time plotter (10), determining if the acquisition is noisy or normal. The process is supported by a logging module (6), a module for managing communication between the operator (3) and internal events (7). In detail, any error committed by the operator (3) or the system itself, whether expected or unexpected, is reported in real time via an abstract user-friendly description (7) and recorded in a SANE private folder (9) with additional information regarding the code location where the exception occurred (6).

Moreover, at the end of each acquisition, always in real time, a partial computation of the data is performed (10), which provides the operator (3) with both the representation of the signals automatically filtered by the disturbances and an estimate of the number of steps that the system can consider valid for each signal. If the amount of data is insufficient for the most important signals, the acquisition is automatically discarded, alerting the operator (7) of the selection’s rationale so that they can make the appropriate adjustments. Thus, the operator is constantly aware of what is happening and is led in their actions so that they can collect complete and consistent data in real time for subsequent offline analysis (12). The described procedure (1) can be performed for several patients and by multiple operators at different times (11), consequently expanding the amount of data recorded in the SANE program directory.

In the operational block (12), the operator (14) can access the data of a certain patient during a specific session (19) in deferred mode, through a new page of their GUI (16). All operations in (16) are monitored and managed by the logging module (17), message box modules (18), and GUI widgets (16). Following the acquisition step (1) comes the offline computational phase (15). Each signal in each section is averaged using a normalized time scale, and within a few seconds, the operator has a complete understanding of the patient and the patient’s average behavior, both scalar and kinematic. The computational offline phase (15) enables a simple, rapid, repeatable, and unaffected by human error monitoring of the patient’s performance. Using these data, the operator, or a medical doctor on their behalf, can independently evaluate the gait analysis’s clinical picture [39].

### 2.2. Description of the System

#### 2.2.1. Tools

Several tools were used to develop the system described in this section. Here is a summary of the elements that were crucial in the realization of SANE. Regarding hardware alternatives, the Intel RealSense d435i RGBD camera was selected due to its cost and performance compatibility with the project’s requirements. The camera d435i is a kind of all-in-one depth sensor with RGB information associated. A stream of information is provided in four channels. The first three channels correspond to the three conventional channels of an ordinary RGB camera (red, yellow, and blue), while the fourth channel represents a matrix of values that can be related to depth (D) [38]. First, depth information is rebuilt by integrating the data from the two stereo camera lenses using standard triangulation techniques. The relative location data of the stereo cameras are already known to the sensor because of a precise calibration performed during assembly. These data are paired with those of an inbuilt IMU inertial sensor, which allows the deduction of the 3D arrangement of the complete camera body. The d435i’s internal PCB automatically captures and processes these data to build an array of depth information that is coherent and well aligned with the RGB pictures. This first level of processing is complemented by information associated with an independent infrared (IR) projector. This data stream is intended to increase the precision of depth information. If stereo cameras were to acquire an image with a pattern that is not particularly recognizable by virtue of relatively homogenous portions of the frame, the channel associated with depth might have difficulty distinguishing features common to cameras and resolving depth data related to the subject. So, the projector irradiates a pseudo-random infrared cloud of points in the acquisition zone, and based on the pattern of the image returned to the sensor, the differences between the predetermined cloud and the irradiated cloud are determined, increasing the detail of poorly defined subjects. IR cloud data are, however, supplementary, and not necessary to produce the RGBD data stream. This decision makes the camera self-sufficient and robust in output, even in the presence of intense sunlight, whose infrared component might disturb the cloud of points. Furthermore, the Intel RealSense d435i is entirely compatible with any image reconstruction software built for deep learning. If the stream already contains well-aligned data, it may be further processed by neural network software to identify more complicated properties shared by the four RGBD channels and make the data more consistent. In conclusion, it is possible to offer a video stream including processed RGB data only, the processed RGBD data, depth image data only, or all this information aligned directly by the sensor without a neural network processing.

Moreover, it was observed that even in absence of infrared projector data, depth acquisition is still feasible. Precisely, highly distinct silhouettes such as the human form remain well distinguished, but the depth matrix in this case would contain unresolved data in the most homogeneous regions such as the ceiling.

On the software side, the AI proposed by Intel has been chosen. It was built by Intel’s partner, Cubemos. It offers the advantage of a very basic Python package that is already prepared to convolve the RGB and D data with the skeleton tracking program and generate an image containing the skeleton of the framed subject along with a map of the three dimensions of each key point of the Cubemos human skeleton model.

The Intel RealSense d435i sensor [38] has been chosen for:The size

The apparatus’s portability makes it extremely compatible with the objective of developing a portable and simple-to-assemble system.
2.Sampling frequencies

They are consistent with the needs of gait analysis in both clinical and sports conditions, as shown below in the section that describes the signal, the signal filtering frequencies, and the design considerations.
3.Costs [40]

The costs are two orders of magnitude lower than those of a laboratory used for gait analysis and one order of magnitude lower than the most modern, portable, and contemporary multi-chamber and lidar systems.

This feature makes the sensor completely suitable for the design requirement of a low-cost system, which is becoming less and less constrained to the clinical setting and has a low economic risk of usage.
4.The ideal range of sampling distance

This property makes the sensor adequate for analyzing a few gait cycles of a person of ordinary height. Nevertheless, the system was also exposed to the tests described below to establish in detail the compatibility of the sensor and the system with the accuracy requirements of the context. Figure 2 shows the reference axes of the Intel RealSense d435i sensor that are used in the system for computation.

#### 2.2.2. Front End

The use of these resources allowed a rapid and synchronous evolution of the graphical interfaces in parallel with the developments of the logic and automation sections (Figure 3). It is important to notice that in the current version of the application, at the beginning the operator had to request the patient to put a certain foot in front of the other and specify it in the interface and indicate the characteristics of the patient’s session, but all these decisions were automated as the work progressed, allowing the patient to be evaluated while walking in a random and natural way.

The aim was both to maximize the intuitiveness of the system for an interface user and to organize the code so that it would be intuitive at first glance too, in case SANE would be further modified in the future.

By virtue of this, the graphic base and stylesheet have been pre-set using the QtDesigner tool and added to the user interface folder and saved respectively in .ui and XML format with Qt widget definitions, their resources, and their private icons, to make the most important modules and those with the most logical contribution easier to understand at the code level.

Each .ui file is imported from the modules responsible for controlling the signals associated with each item in the current widget, offering a second, more abstract level of graphic management and enabling modular and structured code coordination. At this stage, the gait analysis program’s required input controls are implemented.

In addition, dynamic tooltip messages were created in response to the mouse hovering signal and changed each time the program enters a new execution state in order to inform the operator of what SANE expects or why a signal has been disabled. To make the interface more understandable, each icon is also dynamically modified at this point.

The relationship between the interface signals and the functions of all the other control, calculation, and logic packages of the code is handled by a third level of modules designated to control the GUI, which inherits from the modules mentioned above. Therefore, the project main is one of the modules that belong to this level.

Finally, this stage also contains customized modules in the GUI package, particularly the ones that are inherited from the standard PyQt classes and perform similar but not identical functions or those that override some Qt standard methods.

Figure 4 depicts GUI levels from the user’s perspective.

Phase 1: The application opens on the first page of the main window’s QStackedWidget and inhibits navigation. Each button’s tooltip displays a provide folder name alert. The interface is blocked until the patient’s name is submitted in the only available box. If the given name does not comply with the specified format, a new tooltip is created that specifies the input must be alphanumeric.

Phase 2: The start and stop button is now enabled and has a tooltip. When the button is pushed, an Intel RealSense threaded window appears.

Phase 3: The start and stop button is updated both graphically and with a new tooltip so that the acquisition can be terminated when the registered stream is considered sufficient. During this period, skeleton tracking is conducted and Real-Duration Plotter widgets are updated at the same frequency as the video stream, with a maximum data accumulation of five seconds, guaranteeing a lag-free stream for the whole acquisition time, regardless of its length. This uses PyQtGraph-inherited modules.

Phase 4: The save button is activated both graphically and with a new tooltip. The graphs’ tooltips have been updated to facilitate navigating after the acquisition has been completed. The operator can evaluate the quality of the gait based on the “Ankle Distance” signal and the development of the three dimensions of the “Spine” signal (of which the projection in Z can be determined as the approximate distance covered by the patient’s center of gravity towards the camera). A new window is then opened to display on Matplotlib with QtAgg and Navigation Toolbar all the filtered signals and the estimated number of valid steps for each signal.

Phase 5: The operator is now free to repeat the acquisition using the start and stop button, which is subsequently updated, both graphically and with the tooltip, to the state of deletion and restart or saving the acquisition.

In the case of saving, multiple message boxes are shown depending on whether the folder already exists, if there are instances of homonymy, or whether the patient’s name has been modified in an incompatible format.

Phase 6: The data analysis button allows access to the second page of the stacked widget in the main window after several acquisitions.

Phase 7: The page will show a label for entering the patient’s name whose average values are meant to be measured during a particular session, with the same textual controls as the label used in real time, multiple plots Matplotlib with QtAgg and Navigation Toolbar, prepared to display the associated data, and a button that opens a custom QFileDialog so that only one session can be chosen for the patient indicated. Other message boxes are responsible for confirming the folder’s existence, ensuring that its contents are not damaged, analyzing external files, and leading the user through the process.

Phase 8: At any stage, the data acquisition button can be pressed to resume Phase 1.

Lastly, the main window has all the buttons and features of a typical window, including split screen, window minimization, full screen expansion, and, of course, the close button.

#### 2.2.3. Back End

From the point of view of the system logic, one of the most crucial aspects is to provide a sampling time that is sufficient for analyzing each kind of gait, while at the same time considering the demanding computational effort involving an AI and preventing long-duration interrupts from the interface and from concurrent applications that are not part of the system.

As previously mentioned, the multicore features of the Cubemos libraries have been employed to allow the complete usage of the CPU computing power. At the same time, the QThread module of PyQt5 was used to make real-time skeleton tracking independent and asynchronous.

As a result, acquisition and graphical interface loops are disjointed, which improves acquisition robustness and performance.

In the following will be described the module io_hadler.py, which manages signals (inputs and outputs) from the system to the Intel RealSense d435i camera.

In preparation for code production and to establish the optimal parameters, several performance tests were executed. These tests were run in parallel, both on the skeleton tracking module supplied by Cubemos and on the module created for SANE. During this activity, functions from the cProfile library were used to identify the maximum achievable performances, independent of the code written for SANE.

Tests were performed on a desktop computer with an Intel Core i5-4590, 4th Gen, @3.30 GHz CPU and by using the USB 3.1 Gen1 cable supplied by Intel for its sensor. The following assumptions have been made:The system behavior will be similar to the one reported in the following, independently of the hardware on which the SANE software is installed.The system performances will grow linearly and proportionally to the hardware specifics of the test setup.

Additionally, the tests were carried out on hardware whose characteristics were very near to the minimum accepted by the software, as will be justified in the following section. So, it can be stated that our tests were run using hardware that corresponds to the worst-case scenario, where only the bare minimum specs are guaranteed.

Initially, an analysis of each function of the Cubemos package, and of internal calls of the function itself in all modules, was performed by using the cProfile.runctx() function: the outcome was a series of comma-separated values (CSV) files, one for each test. As described in [41], the CSV file includes the name of the task, the line of code and the modules in which the function call happens, the total number of function calls, the total time per call (in milliseconds), and several other data useful for analysis of performances. An Excel file was obtained from the described CSV to opportunely sort data.

From this information, it can be concluded that the wrapper.py module can be considered the bottleneck for AI skeleton tracking. Indeed, wrapper.py contains the primary function used for key point estimation, which is called iteratively during the analysis of each video frame.

When performing a 100-call, 1 frame per call test, the time spent for each call averages 64 ms. By also considering the performance lag caused by the cProfile queue, we can conclude that the maximum number of frames per second (FPS) that SANE is able to provide on our hardware, with the source code offered by the Cubemos API, can be approximated by default in Equation (1):(1)$$\frac{100}{64}\mathrm{ms}=\frac{1s}{\;T_{bottleneck}}=f_{max\;}=15,625\;\mathrm{Hz}$$

The above test used the cProfile library and can be considered useful for the estimation of bottlenecks since we can evaluate the time employed by each function in relation to the others. However, these kinds of tests are less reliable when it comes to the evaluation of function timing in an absolute way since the tests themselves involve the presence of a non-negligible computational effort.

As a result, additional frequency tests will be performed on SANE and on the Cubemos skeleton-tracking-realsense.py module. These tests will make use of the datetime library and will involve a smaller computational effort since the scope of the activity is to assess how much time it takes to return each video frame after it was submitted for AI analysis.

In detail, the results depend on both the estimation time of AI and on the code structure, and they show the accumulated delay on all the code when performing a frame-by-frame analysis of the video.

From the described assessment, it appears that the skeleton tracking implementation proposed by Cubemos guarantees an average data frequency of 12 Hz, which is 3 less than the ones guaranteed by the pure AI point extraction computed with cProfile.

As previously mentioned, the sampling frequency is critical for obtaining a good gait analysis. Therefore, the direct usage of the Cubemos module was discarded. The alternative choice was the adjustment, whenever possible, of the Cubemos Python package, followed by the integration of the changes into SANE. The procedure was carried out with as little impact on the pure AI analysis times as possible.

Therefore, file io_handler.py contains the code provided by Cubemos for the image analysis, but transposed in form of class, adapted to the PyQt environment, and optimized for real time and in thread.

An initialization section can be found in the object created by the io_handler.py class. The initialization contains preliminary settings for license verification, resolution configuration, settings for the analysis confidence of the tracker, graphic settings for the acquisition window, and configurations of camera stream alignment and the frequencies needed for the depth and RGB channels.

In particular, the initialization section of the object created by io_handler.py is called by main.py at the interface start-up, and it is kept on for the entire duration of the main window loop. This, specifically, allows moving at the beginning all the delays caused by the functions that need to be called at the program start-up. Since all lags have been shifted at the beginning, this allows an immediate response from the skeleton tracker: operators are able to acquire data exactly when they intend to, start the acquisition, and cue the patient at the beginning of the walk recording.

In accordance with the Intel RealSense guidelines for the Python environment, the acquisition is set to provide both RGB and D data. Both channels have a common maximum frame frequency set to 60 FPS [38]. This is useful to avoid software bottlenecks that would happen if the SANE code were run on a computer with high computational performance hardware, able to resolve skeleton tracking in less than 0.01 s.

The resolution has been adjusted to the highest possible setting, at 640 × 480 px. This selection is the optimal choice in terms of performance—that is, it does not affect the skeleton tracker maximum frame rate—and was obtained by analyzing all possible combinations of frequency and resolution. The activity was conducted using the cProfile library testing functions.

Once alignment of frames and pipeline initialization are completed, the system is ready to be activated by the creation of the recording window. Lastly, the run() method of the thread contains the final part of the code, whose purpose is the generation of a video where skeleton tracking is shown.

The thread includes the pipeline start, checks to verify the correct alignment of frames, and queues new frames characterized by a time limit needed to signal connection issues with the camera. The thread also includes the 2D skeleton tracking functions and the function to extract 3D data based on 2D coordinates. For each frame, the following items are sent to the modules called by io_handler.py:Two timestamps, one for the elapsed time between two consecutive frames and the other for the time elapsed from the beginning of the acquisition. Those are obtained by using a custom class derived from the built-in QTimer of PyQt.A list of the 3D skeleton key points in the considered frame.The processed image.

The items are to be displayed via QLabel and PyQtGraph. Then, new real-time data with negligible computational cost are calculated, and real-time data are saved.

Good efficiency levels can be obtained by implementing the real-time management of plots as described in previous paragraphs of this paper: image update ensues when a frame is available, which only involves changing the QPixmap of the QLabel that simulates the video. In the meantime, csv_handler.py generates the .csv file containing the real-time data. The file is opened and left in append when the video stream begins, and it will be either closed at the stop of the video or deleted if any unexpected interruption occurs.

If a key point is not provided by the AI at the time of frame-by-frame addition of new data, the csv_handler.py is designed to generate NaN data; in addition, csv_handler.py produces six new signals thanks to the support functions from the computation_handler.py module:1.The signal related to the “Pelvis” key point:It is computed as the midpoint between the “Right Hip” and “Left Hip” signals, is essential to compute several angles in gait analysis, and is a common reference point in motion tracking literature. Despite this, Cubemos does not provide this point, which is therefore computed at this moment. It consists of three further signals, which are the projection of the “Pelvis” on the *X*, the *Y*, and the *Z* axes.2.The “Ankle Distance” trend:It is a fundamental signal for gait recognition. It can be computed as the distance between the projection on the *Z* axis of the Left Ankle and the Right Ankle signals, specifically by subtraction of the former from the latter. Each time a gait cycle is performed, regardless of unsteadiness or asymmetry of the stride, the Ankle Distance signal has a sine-like shape with a complete period and with variable amplitude.3.The Total Displacement signal:It coincides with the range (R) defined by Intel RealSense, and it can be computed using Equation (2):(2)$$\sqrt{SpineX^{2}+SpineY^{2}+SpineZ^{2}}\;$$
where the addendums are the 3 dimensions of “*Spine*”, which is the key point commonly considered in the literature as the reference for the movement of the whole skeleton.4.The FPS:It is computed as the reciprocal of the period between two frames, where standard numerical controls for real-time division must be employed.

All the above-mentioned parameters can be considered acceptable only if all the events that generate them result in valid outcomes. That is, if any of them equals NaN, all computations for that particular instant are stopped and a NaN solution is given.

Various tests have been performed employing both the cProfile and the datetime libraries. Tests consider cases with and without the computations performed by csv_handler.py, with and without the portion of code responsible for time plot and video display. Results show that, overall, the FPS is not affected by these variations. Even though the sections of code considered have a total execution time that is strictly dependent on the number of key points, the execution time of wrapper.py, used by the AI, is almost two orders of magnitude bigger. Hence, multithreading was abandoned, and the solution of leaving the original modules running sequentially with respect to each thread cycle was preferred. However, it should be noted that, given high computational hardware, the multithreading solution would have been a reasonable solution also able to: use an asynchronous sample of the data, acquired at 30 FPS, to reduce video and plot updates; save numerical data only at the highest possible frequency, sequentially, and synchronously; and immediately release a frame analyzed by the AI, so that new ones, still in the pipeline, can be processed. These limited benefits were not considered sufficient to justify such an unsafe design choice.

Based on execution reports provided by cProfile, it was assessed that, in the best-case scenario, a representation time improvement for each frame of slightly above 1% could be obtained. Still, this minor improvement would have imposed constraints on the project, making several parameters not customizable. One example is the time interval of the plot that can be represented in real time. As the data to represent for each frame grows, the risk of making the output thread slower than the input thread increases. This would result in asynchrony, inconsistencies between plots and images, and data accumulation, with the added possibility of memory consumption fault. It must also be taken into account that the d435i’s camera is able to deliver images at a maximum rate of 60 FPS. So, even if the Cubemos AI were faster, the 60 FPS limit would still exist. In addition, this kind of frequency for a video and plot representation is reasonable, as it is well within the limits obtained by applying the Nyquist criterion to the sampling rate of the human eye.

The end of the acquisition is associated with a final real-time check on the quality of data. Function check_signal_minimal_requirements(), present in the computation_handler.py module, is called. The function evaluates the complete dataset and states if the acquired gait can be used to compute average values. It also displays the number of steps and signals correctly recognized, which present a valid dataset within the period of the steps. In case of bad acquisitions (total number of recognized steps equal to zero), the operator is prompted to repeat the gait analysis and the following information is shown:The signal is not shaped in the form of a gait cycle: the number of steps is zero because the patient is not able to execute all the gait phases.The acquisition is non-continuous because the patient stopped moving during each cycle: the number of steps is zero since data are not suitable for analysis.The signal is not reliable either because too much noise is present in the video or because the AI outputs too many NaN results.The sampling time fell below the minimum 16 FPS threshold. This happens when the video is recorded on obsolete hardware or if the PC running SANE is executing too many programs simultaneously. Moreover, if some key points in every step have produced signals affected by noise, the operator receives a warning: the system is able to compute reliable averages provided that at least one of the acquisitions contains acceptable data of that signal. The operator is also warned if too much noise affects the acquisition.

In its final configuration, SANE proved it could analyze the 3D skeleton with frequencies between 16 and 18 FPS, on average 17.2 FPS on a 10-walk test. That is six more FPS than the value obtained with the Cubemos example. Additionally, if compared to the maximum frequency estimated via cProfile, SANE could achieve a higher frequency. All the evidence indicates that the system is able to deliver ideal performances, close to the optimal real-time ones.

In the following, a top-down description of the system logic will be provided, defining mathematical criteria that allow both system automation and the definition of minimum design requirements for robust analysis.

As concerns data calculation processes, they are conducted offline. Their final goal is to provide:Scalar data such as speed, step duration and length, cadence, and cycle duration. These statistics are averaged over all the steps that are present in all the acquisitions of a single session.Kinematic data, specifically data about the variation in time of the angles associated with the step. These signals are averaged and normalized over all the steps that are present in all the acquisitions of a single session. The angles include obliquity, tilt, and rotation of the trunk; pelvic tilt and rotation; hip abduction/adduction and flexion/extension; the varus–valgus and flexion/extension of the knee. The signals for pelvic obliquity, knee and hip rotation, ankle dorsi-plantarflexion, and foot progression are not present since the points provided by Cubemos are not sufficient for mathematically defining these angles and recognizing body sections as the feet or the hands.

This information can be obtained from the CSV file of the acquisition, which can be found in the session folder. The mean values of these acquisitions are then saved into the parameters of an object created by the class of the computation_handler.py module along with the total number of valid steps. This makes it possible to evaluate the reliability of the measurements.

In the end, a vector for each parameter is obtained for each walk, together with its average and standard deviation. A normalization over time of the signals is performed to adjust them to a common range. The scale on which normalization is performed ranges from 0 to 100. The involved data are all the signals associated with every step in each CSV file. Resampling of the variables to a common frequency is also applied. This final resampling frequency is computed based on all session acquisition frequencies.

In order to obtain the discussed signal characteristics, an iterative analysis of the CSV file and an average of all signals must be performed. So, the basic info to store in the CSV file is the number of steps correctly acquired and the acquisition time.

Then, the criteria to define a “good signal” must be established in terms of step recognition and timing.

For instance, in order to determine the list of times when a given angle began and ended its period during an entire acquisition, it is necessary to have the same data for the x, y, and z signals associated with the points generating the angle, then determine which valid time spans in this list are shared by all these points, and finally define the list of common time spans as the vector of information sought for these angles.

In terms of code, all the signals of an acquisition are saved in a data frame as a time series. From each of these first matrices, we can derive a second matrix, whose elements are Boolean values that indicate whether the element in the first matrix in the same position can be considered valid or not. Each NaN entry in the first matrix corresponds to a False entry in the second matrix. Any outlier entry in the first matrix corresponds to a False entry in the second matrix.

This easy process justified the existence of this mask matrix, which is a temporary structure that occupies additional memory.

Disturbances and outliers are also removed by filtering the signals with an IQR filter that automatically generates another mask vector for all the signals that are not within the 25th–75th percentiles and are therefore not valid.

The superposition of this information with the position of the NaN signals allows obtaining the complete mask matrix.

Then, a time alignment is needed. A time alignment is the recognition of contemporary events in the time series vector signals, precisely in the X, Y, and Z projections of a key point. So, it is necessary to identify the portions that are simultaneous with the event of a recognized step by using the Ankle Distance trend, and then, based on the distribution of True and False values in the section of the associated mask matrix, determine whether that range of values is a good step for the current signal.

Specifically, in each signal section contemporary to the cycle of a step, the maximum train of consecutive False is identified, its elements are counted, and it is determined whether this number is less than or equal to 30% of the total number of points required to reconstruct the signal’s highest frequency component, given the average sampling frequency in real time.

Automatic step detection uses Ankle Distance, which has a sinusoidal-like shape. In order to recognize a complete cycle, it is sufficient to select the points in which the signal crosses the 0 value and search for successive sequences of absolute max–min–max or min–max–min points. The sets where the patient’s speed is not continuous are discarded, along with the ones where the patient is standing still (zero speed) or walks away from the acquisition (negative speed), as shown in Figure 5 and Figure 6.

### 2.3. Statistical Analysis

Multiple tests were conducted on two different occasions in the Gait Analysis Laboratory of the Clinical Research Centre Neuromed in Pozzilli, IS, in order to validate the results obtained from SANE.

The site is equipped with all the instruments required by the latest standard BTS GAITLAB protocol, except for the number of dynamometric plates, which in this case, amounts to two. However, the latter were simply employed to simulate a complete analysis process and have no relevance to the evaluation and comparison of kinematic and scalar data between BTS and SANE. Dynamic data will thus not be discussed in this section dedicated to comparison tests, since they are not supplied by SANE. Consequently, four sessions of 10 acquisitions were performed by two distinct operators on the same healthy patient in various modalities, for a total of 40 acquisitions by SANE and 40 simultaneous acquisitions by BTS used as the gold standard.

More specifically, the procedures shown in Table 3 have been followed:Inter-rater test: Two different operators perform the entire BTS marker placement procedure for the acquisition of two consecutive sessions, one each, of 10 walks, using SANE and BTS simultaneously on the same patient on the same day. The aim of these tests is not only to provide results from the two systems, but also to show the operator-induced measurement error between the two systems. In other words, it is an estimate of the systems’ dependency on the operator, as well as their reliability in this regard.In the context of this study, session 1 and session 2 were therefore associated with this test, performed by both operator 1 (Eval 1) and operator 2 (Eval 2)Test–retest: It is executed by the same operator who performs, on the same patient, on two consecutive days, two days apart [44], ex novo, the whole process for the placement of the markers predicted by the BTS to acquire two discontinuous and remote sessions of 10 walks using BTS and SANE concurrently. The aim of these tests is to demonstrate the difference between measurements collected in different periods by the same operator and patient. In other words, it is an evaluation of the systems’ dependence on time, as well as their reliability in this regard.In the context of this study, session 2 and session 3 were therefore associated with this test, performed by operator 2 (Eval 2).Intra-rater test: It is done by the same operator who performs, on the same patient on the same day, ex novo, the same process for the installation of the BTS-prescribed markers, for the acquisition of two consecutive sessions of 10 walks, using BTS and SANE concurrently. These tests are meant to demonstrate the variation between repeated measurements performed by the same operator on the same patient.In the context of this study, session 3 and session 4 were therefore associated with this test, performed by operator 2 (Eval 2).

Sessions 1 and 2 were conducted on the same day, as were sessions 3 and 4. The latter were separated by two days from the former.

For optimal acquisition, gait analysis was performed in the center of the capture area of the BTS laboratory, with the Intel RealSense d435i sensor positioned along the patient walk trajectory, which was three meters long and centered on the dynamometer plates.

The RealSense camera was positioned on a tripod 1.63 m from the end of the patient’s path, 1.30 m above the floor, and at an angle of +13° along its *X*-axis and 0° along the other axes, with the patient’s planned trajectory visible. In order to avoid any form of light interference, the floor was marked with nonreflective tape. The room is shielded from sunlight to prevent infrared disturbances, and all objects from the acquisition zone were eliminated in order to have a completely empty chamber (Figure 7).

The placement of the markers followed the Davis protocol.

This configuration remained unaltered for the duration of all experiments.

Figure 7 shows the complete layout of the experiment. It includes (a) part of the infrared cameras required by the BTS for acquisition. They are placed on all 4 sides of the room for a 360° acquisition. The layout includes (b) a section of the floor occupied by the dynamometer plates. This also generally corresponds to the section of space for optimal acquisition in a room equipped with BTS systems. The layout includes (c) the planned acquisition trajectory, marked with high-contrast non-reflective tape to indicate the direction of travel to the patient. It intersects with both the dynamometric section of the BTS and its optimal acquisition area. It extends towards the SANE sensor. The length and distance of the path allow optimal acquisition for SANE as well. The layout includes (d) side-by-side monitors of SANE and BTS to allow synchronous measurement start-up and supervision by the same operator. The layout includes (e) SANE’s Intel RealSense camera on top of its tripod in the patient’s direction of travel.

## 3. Results

The tables of results are shown below.

Each of the scalar values measured by both BTS and SANE has its own table. In each table, each row represents one of the three types of tests described.

For ease of reading, starting from the left-hand side, the first column of each row has been associated with a second subset of rows indicating the session and, thus, the dataset from which the results were derived. The second and third columns, respectively, reflect the BTS and SANE values. The latter are further separated into two sub-columns indicating, in order, the mean and standard deviation values computed by the two systems for each session, and the relative mean percentage error (REM%) value, calculated from each test for each measurement using the following Equations (3) and (4):(3)$$REM_{SANE}\%=\left|\frac{\bar{x_{Session\_SANE_{i+1}}}-\bar{x_{Session\_SANE_{i}}}}{\bar{x_{Session\_SANE_{i}}}}\right|\times 100\;$$
(4)$$REM_{BTS}\%=\left|\frac{\bar{x_{Session\_BTS_{i+1}}}-\bar{x_{Session\_BTS_{i}}}}{\bar{x_{Session\_BTS_{i}}}}\right|\times 100$$

### 3.1. Gait Cycle Duration

It is possible to notice similar average and standard deviation results for the gait cycle duration. Regarding Table 4, in the inter-rater relative error measurement and test–retest relative error measurement, SANE and BTS had comparable results. In the inter-rater test, the REM% of SANE was 8.27% and BTS was 7.41%, while in the test–retest, the REM% value was 4.17 for SANE and 4.83% for BTS. In the intra-rater test, the REM% was 1.45% for SANE and for 3.62% BTS, and the difference between SANE and BTS was 2.17%.

### 3.2. Gait Step Duration

Regarding Table 5, in the inter-rater test, the average relative error in the BTS system had a value of 22.86%, while in SANE it had a value of 9.09%; in the test–retest, BTS had a value of 5.81% while SANE had a value of 4.17%. In the intra-rater test, SANE exhibited a REM% of 1.45% while BTS exhibited a value of 4.94%.

### 3.3. Gait Cadence

Regarding Table 6, it is possible to notice similar average and standard deviation results for the gait cadence. The inter-rater test value for SANE was 8.59%, comparable to the one obtained with the BTS system, 8.09%. In test–retest, the BTS had a value of 5.51% while SANE had a value of 4.40%. However, the values are still similar between the two systems.

### 3.4. Gait Cycle Length

Regarding Table 7, in the case of the gait cycle length, SANE performed with a REM% in the inter-rater test with a value of 3.57% while BTS exhibited a 1.50% REM%. In the test–retest, the REM% of SANE was 0.69%, while BTS reached a value of 5.93%. The intra-rater test results are comparable between the two systems, with 2.74% for SANE and 2.80% for BTS.

### 3.5. Gait Velocity

Regarding Table 8, in the case of the gait velocity, SANE performed with a REM% in the intra-rater test of 3.77%, while in the inter-rater test and the test–retest, the BTS REM% values were 10.00% and 11.11%, respectively. However, these measurements in the case of the BTS system present a null standard deviation in five sessions out of six.

## 4. Discussion

During the execution of the tests, BTS vulnerabilities emerged. Phantom markers emerged sporadically, and although some of the acquisitions appeared to be accurately recorded in the 3D visual feedback window, they were not pickable during the offline phase.

Following the meticulous selection and identification of 20 right stances, 20 left stances, 10 right detachments, and 10 left detachments over three sessions and a total of 180 manual signal choices using the 3D window supplied by the BTS Smart Clinic program, the manual gait selection procedure was completed.

The offline analysis program had occasional crashes too, although none occurred during the period of real-time data collecting, thus not affecting the measurements, while only causing delays and possibly patient stress.

During the real-time phase, based on the recording of the 3D skeleton created in real time by the markers, data were recollected with both systems whenever the BTS dataset was visibly altered using the basic feedback tools of this system.

The SANE system, on the other hand, requires the reacquisition of the walks just twice, once in conjunction with the BTS. In order to maintain the comparability of the tests, the acquisition process was repeated with both systems in this case too.

During the offline phase, the walks that were eliminated by one system were not also discarded in the other, since the comparison of the two methods is limited to the real-time acquisition operations.

Therefore, after selecting the steps, the BTS consistency analysis phase for Session 1 of operator 1 required discarding acquisitions 01, 06, and 07; for Session 2 of operator 2, acquisitions 02 and 03; for Session 3 of operator 2, acquisitions 01, 06, 07, 08, and 09; and for Session 4 of operator 2, acquisitions 01, 03, 04, 05, 06, 09, and 10.

It is essential to note that none of them were required to be eliminated by the SANE system.

The averages thus obtained are the ones shown in Section 3. According to what can be deduced from them, the findings are promising.

SANE has a lower relative percentage error than BTS in all test–retest comparisons, less than 5%, with the highest value of 4.95% when computing the average speed, compared to the BTS maximum of 11.11%. This is the first indication that SANE exhibits higher repeatability.

In terms of intra-rater testing, SANE consistently produces highly favorable data, always less than 5% REM% and always lower REM% values than those of BTS, except for the average speed table. In addition, this very last average provided by the BTS is implausible for two reasons: the BTS’s tendency to truncate too many digits of the standard deviation, frequently declaring an implausible standard deviation of 0, and the BTS’s smaller number of acquisitions in sessions 3 and 4, which were discarded for the reasons discussed in Section 3. Meanwhile, the same dataset was completely averaged using SANE, providing a low but non-zero standard deviation.

In the inter-rater tests, SANE returned a better value in terms of REM% than BTS, up to a gap of 7%, and a slightly higher value than BTS, making the measurements comparable between both the two systems, with a maximum gap of 0.5%, 0.86%, and 2.07%. SANE, on the other hand, has a maximum REM% of 9.09%, whereas BTS has a maximum REM% of 22.86%. This third term of comparison leaves margins for interpretation.

The mean and standard deviation measures of SANE are comparable to those of BTS.

While it is difficult to determine which of the two systems returned the most accurate measurement in terms of absolute value, given the nature of a human step, it is much easier to evaluate both systems’ repeatability and reliability, emphasizing that the latter evaluation is more important than the numerical accuracy of the measurement. After all, while an incorrect measurement from a repeatable system may be fixed by calibration, a fortuitously accurate result from a non-repeatable system can never be guaranteed to be reliable or improved in any manner.

If SANE’s results were to be considered comparable with those of BTS, this would still turn in favor of SANE given its convenience in several other aspects such as its speed, compactness, automation, lower price, and lower stress.

As a result of what has been reported, SANE has been proven to have more repeatability and low exposure to human influences in various instances, providing justification for conducting further future tests on the system to confirm this claim.

## 5. Conclusions

As shown by the study’s findings, SANE demonstrates that it is an effective method for gait analysis. As discussed in further detail below, what was lost in accuracy owing to the use of AI and a frontal camera with a smaller sampling rate than stereophotogrammetry systems is returned due to the automation, ease of use, high repeatability, and removal of human error.

Since the patient is not needed to undergo lengthy marker preparation and wearing procedures, the technique has been demonstrated to be much less stressful. SANE looks to be an exceptionally beneficial methodology for studying the gait-impaired persons who cannot physically support the preliminary procedures for marker analysis, providing new options for a segment of the population who previously had no access to this sort of study.

During the experiments conducted at the Clinical Research Centre Neuromed in Italy, a significantly lower rate of acquisition failure has been reported compared to the gold standard methods, owing to the reduction in the number of variables that may impact the measurement while walking. The system’s robustness is also attributable to its real-time feedback and self-debugging capabilities, which enable it to recognize the existence and type of a problem during the acquisition process.

Another significant feature of the system is its portability, which enables acquisitions to occur in non-clinical settings. Additionally, the system’s high degree of automation and entirely guided interface allow untrained operators to use it, substantially reducing costs and system restrictions. As a result, the benefits include a significant improvement in terms of time and cost, which are two orders of magnitude less expensive than those associated with the BTS.

However, the path to the present result was not without setbacks and obstacles. The usage of Cubemos as artificial intelligence still imposes certain constraints on the system. A fraction of BTS’s angles cannot be analyzed using SANE owing to Cubemos’ lack of sufficient points. Additionally, Cubemos software highlighted issues with GPU use, making the code developed for SANE potentially more promising in terms of performance on the one hand, but still constrained by external modules on the other. Cubemos has not released any software upgrades in recent months, and its relationship with Intel has deteriorated. Recently, Intel has also chosen to cease its focus on sensor technologies such as the RealSense camera, jeopardizing the project’s future.

That is why SANE has been consequently made very scalable, flexible, and modular during the latter stages of development, enabling it to be used with different types of AI and acquisition tools with just slight alterations.

The system’s future objectives include evaluating potential enhancements, incorporating new AIs, recognizing patterns other than skeletal tracking, adapting SANE to analyze other regions of the body, identifying pathway geometries, and in potential long-term projects recognizing the patient’s emotional response during therapy. The possibilities for teaching, deploying, and enhancing AI are nearly endless.

All in all, SANE finds a natural and fitting application in the area of preliminary gait analysis systems to obtain rapid feedback on a patient’s features and determine whether to perform additional in-depth assessments.

## Figures and Tables

**Figure 1 ijerph-19-10032-f001:**
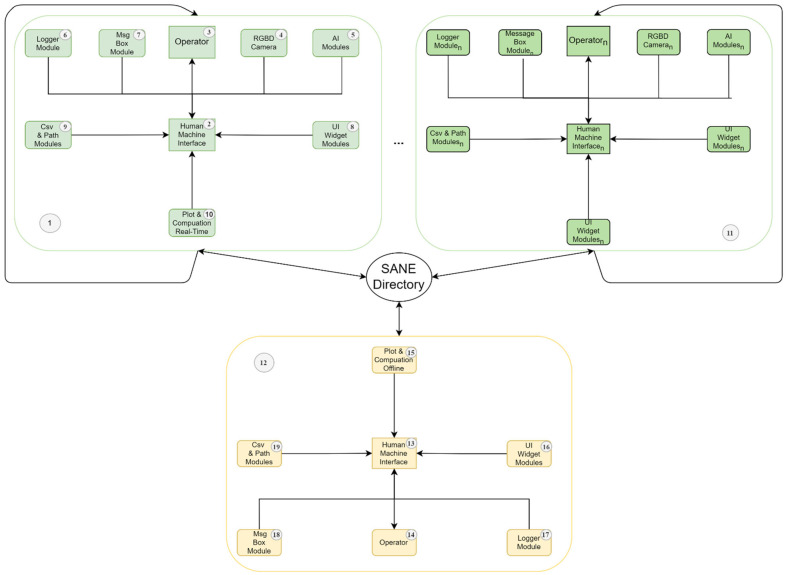
Proposed methodology.

**Figure 2 ijerph-19-10032-f002:**
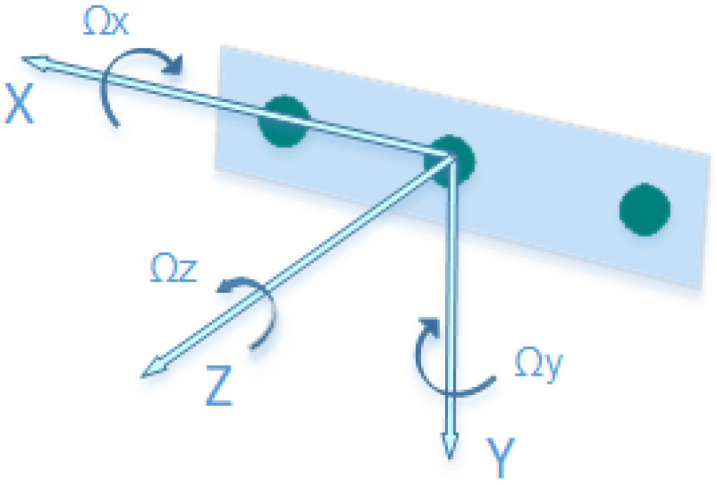
Intel RealSense D435i reference axes [38].

**Figure 3 ijerph-19-10032-f003:**
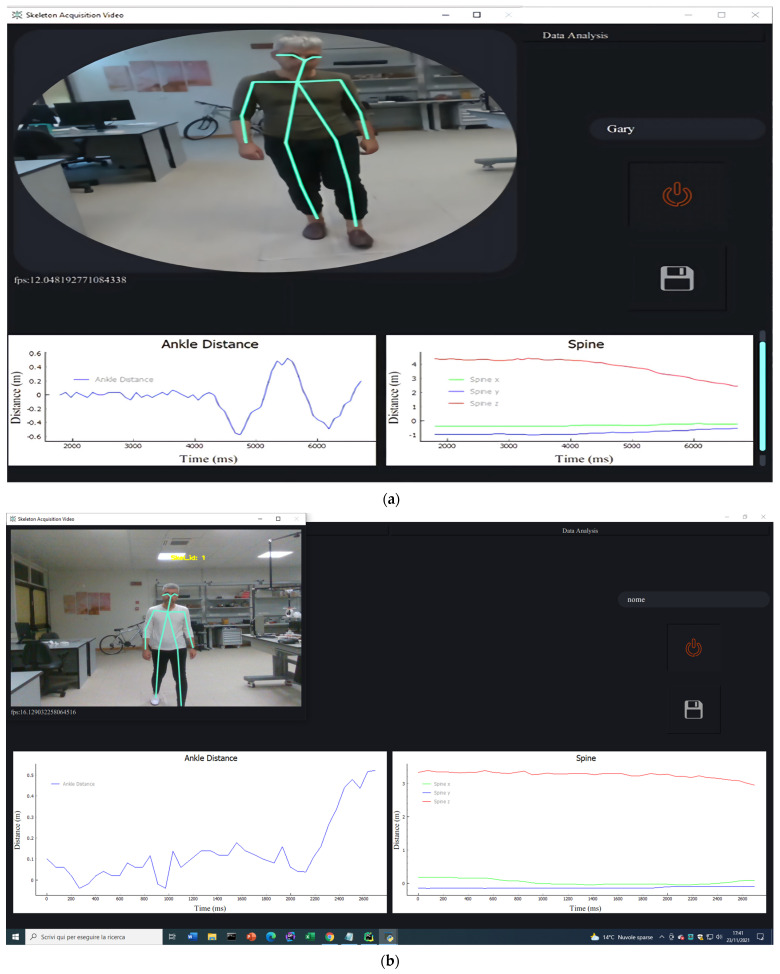
(**a**) SANE’s second prototype at the end of the acquisition of a step. (**b**) Actual SANE’s GUI at the very beginning of a gait cycle acquisition with improved acquisition speed.

**Figure 4 ijerph-19-10032-f004:**
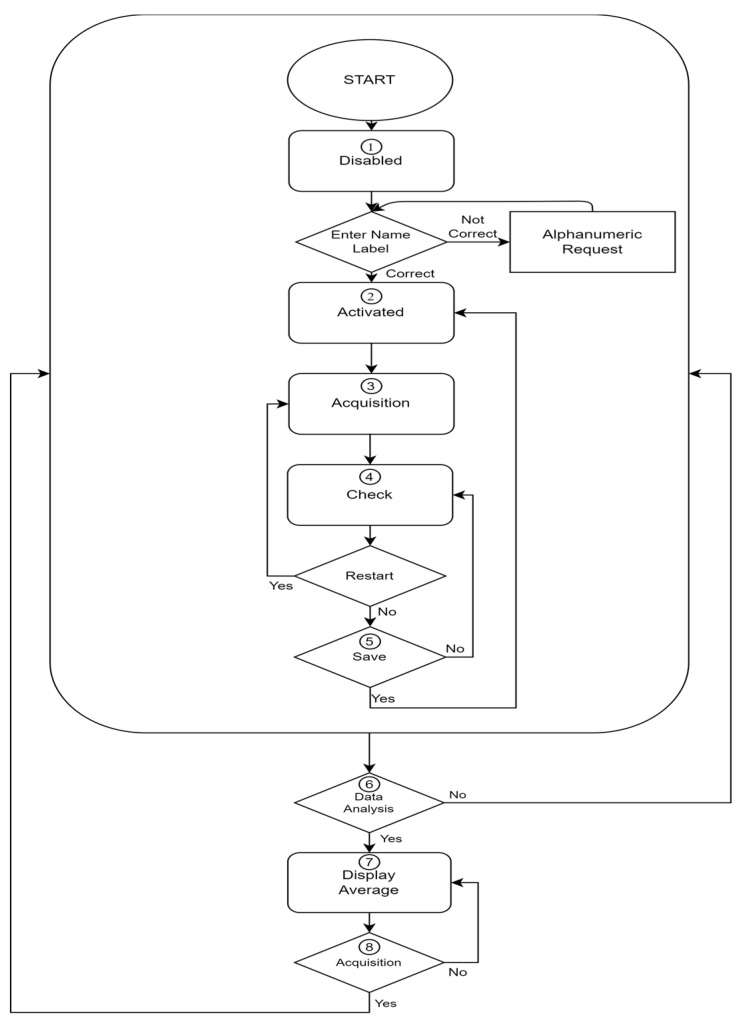
Operator interface flow chart.

**Figure 5 ijerph-19-10032-f005:**
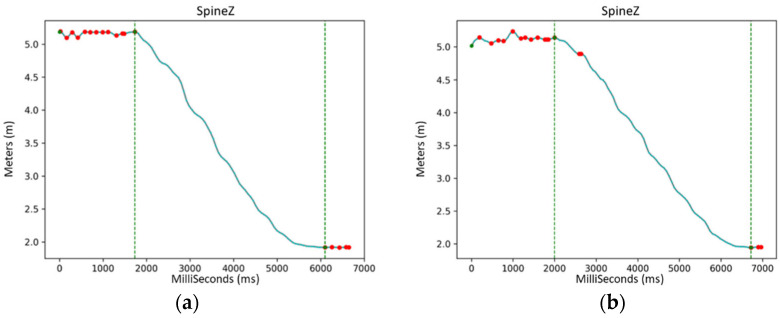
SANE’s detection of movement’s start and stop (green vertical lines) through the projection on Z of the signal related to the Spine joint and its relative maxima and minima (red dots): (**a**) a low-noise acquisition while walking; (**b**) a different, slightly noisier acquisition with a maximum and a minimum that are correctly ignored during the gait recognition and the start and stop evaluation.

**Figure 6 ijerph-19-10032-f006:**
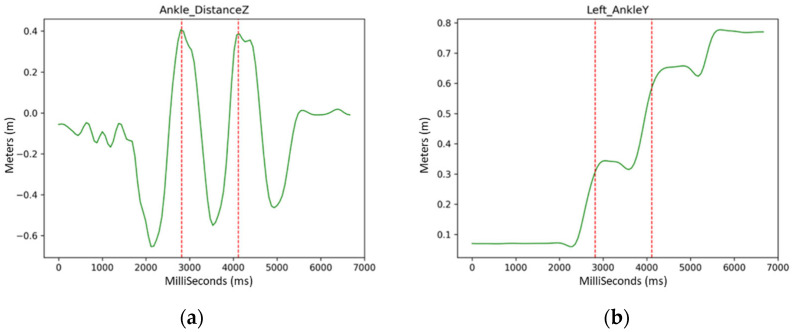
SANE’s gait detection (red vertical lines) via the projection on Z of the trend related to the Ankle Distance: (**a**) Ankle Distance trend; (**b**) an example of a key point signal (the projection on X of the signal related to the Right Ankle joint) cut within the time range associated with the gait detection.

**Figure 7 ijerph-19-10032-f007:**
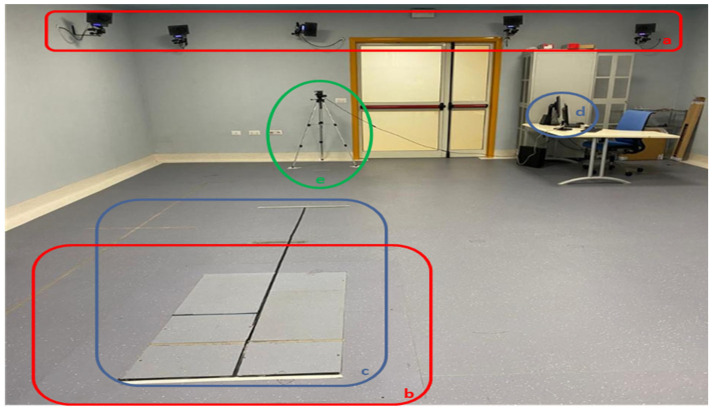
Testing layout.

**Table 1 ijerph-19-10032-t001:** Comparison of optical systems based on depth measurement [23].

Method	Advantages	Disadvantages	Each Sensor Price (EUR)	Ref.	Accuracy
Camera Triangulation	High image resolution	- At least two cameras needed - High computational cost	400 to 1900	[24,25]	70% [25]
	No special conditions in terms of scene illumination				
Time of Flight	Only one camera is needed	- Low resolutions - Aliasing effect - Problems with reflective surfaces	239 to 3700	[23]	2.66% to 9.25% (EER) [23]
	It is not necessary to calculate depth manually				
	Real-time 3D acquisition				
	Reduced dependence on scene illumination				
Structured Light	Provides great detail	- Irregular functioning with motion scenes - Problems with transparent and reflective surfaces - Superposition of the light pattern with reflections	160 to 200	[26,27]	<1% (mean diff) [27]
	Allows robust and precise acquisition of objects with arbitrary geometry and a wide range of materials				
	Geometry and texture can be obtained with the same camera				
Infrared Thermography	Fast, reliable, and accurate output	- Cost of instrument is relatively high - Unable to detect the inside temperature if the medium is separated by glass/polythene - Emissivity problems	1000 to 18,440	[28]	78–91%
	A large surface area can be scanned in no time				
	Requires very little skill for monitoring				

**Table 2 ijerph-19-10032-t002:** Non-wearable system (NWS) and wearable system (WS) comparison [23].

System	Advantages	Disadvantages
NWS	- Allows simultaneous analysis of multiple gait parameters captured from different approaches - Not restricted by power consumption - Some systems are totally non-intrusive in terms of attaching sensors to the body - Complex analysis systems allow more precision and have more measurement capacity - Better repeatability and reproducibility and less external factor interference due to controlled environment - Measurement process controlled in real time by the specialist	- Normal subject gait can be altered due to walking space restrictions required by the measurement system - Expensive equipment and tests - Impossible to monitor real-life gait outside the instrumented environment
WS	- Transparent analysis and monitoring of gait during daily activities and in the long term - Cheaper systems - Allows the possibility of deployment in any place, not needing controlled environments - Increasing availability of varied miniaturized sensors - Wireless systems enhance usability - In clinical gait analysis, promotes autonomy and active role of patients	- Power consumption restrictions due to limited battery duration - Complex algorithms needed to estimate parameters from inertial sensors - Allows analysis of limited number of gait parameters - Susceptible to noise and interference of external factors not controlled by specialist

**Table 3 ijerph-19-10032-t003:** Session plan.

	First Day	Two Days Later
**Eval 1**	Session 1	
**Eval 2**	Session 2	Session 3 Session 4

**Table 4 ijerph-19-10032-t004:** Results for gait cycle duration.

		SANE		BTS System	
		Mean ± SD (s)	REM%	Mean ± SD (s)	REM%
Inter-rater relative error measurement	Session 1	1.33 ± 0.10	8.27	1.35 ± 0.11	7.41
	Session 2	1.44 ± 0.07		1.45 ± 0.04	
Test–retest relative error measurement	Session 2	1.44 ± 0.07	4.17	1.45 ± 0.04	4.83
	Session 3	1.38 ± 0.06		1.38 ± 0.04	
Intra-rater relative error measurement	Session 3	1.38 ± 0.06	1.45	1.38 ± 0.04	3.62
	Session 4	1.40 ± 0.06		1.43 ± 0.03	

**Table 5 ijerph-19-10032-t005:** Results for gait step duration.

		SANE		BTS System	
		Mean ± SD (s)	REM%	Mean ± SD (s)	REM%
Inter-rater relative error measurement	Session 1	0.66 ± 0.05	9.09	0.70 ± 0.12	22.86
	Session 2	0.72 ± 0.04		0.86 ± 0.02	
Test–retest relative error measurement	Session 2	0.72 ± 0.04	4.17	0.86 ± 0.02	5.81
	Session 3	0.69 ± 0.03		0.81 ± 0.02	
Intra-rater relative error measurement	Session 3	0.69 ± 0.03	1.45	0.81 ± 0.02	4.94
	Session 4	0.70 ± 0.03		0.85 ± 0.02	

**Table 6 ijerph-19-10032-t006:** Results for gait cadence.

		SANE		BTS System	
		Mean ± SD (Step/min)	REM%	Mean ± SD (Step/min)	REM%
Inter-rater relative error measurement	Session 1	91.48 ± 8.94	8.59	90.09 ± 5.24	8.09
	Session 2	83.62 ± 4.30		82.80 ± 2.29	
Test–retest relative error measurement	Session 2	83.62 ± 4.30	4.40	82.80 ± 2.29	5.51
	Session 3	87.30 ± 3.57		87.36 ±2.34	
Intra-rater relative error measurement	Session 3	87.30 ± 3.57	1.10	87.36 ±2.34	3.85
	Session 4	86.34 ± 3.98		84.00 ± 1.77	

**Table 7 ijerph-19-10032-t007:** Results for gait cycle length.

		SANE		BTS System	
		Mean ± SD (m)	REM%	Mean ± SD (m)	REM%
Inter-rater relative error measurement	Session 1	1.40 ± 0.17	3.57	1.33 ± 0.16	1.50
	Session 2	1.45 ± 0.11		1.35 ± 0.04	
Test–retest relative error measurement	Session 2	1.45 ± 0.11	0.69	1.35 ± 0.04	5.93
	Session 3	1.46 ± 0.07		1.43 ± 0.03	
Intra-rater relative error measurement	Session 3	1.46 ± 0.07	2.74	1.43 ± 0.03	2.80
	Session 4	1.42 ± 0.08		1.39 ± 0.02	

**Table 8 ijerph-19-10032-t008:** Results for gait velocity.

		SANE		BTS System	
		Mean ± SD (m/s)	REM%	Mean ± SD (m/s)	REM%
Inter-rater relative error measurement	Session 1	1.05 ± 0.08	3.81	1.00± 0.10	10.00
	Session 2	1.01 ± 0.07		0.90 ± 0	
Test–retest relative error measurement	Session 2	1.01 ± 0.07	4.95	0.90 ± 0	11.11
	Session 3	1.06 ± 0.07		1.00 ± 0	
Intra-rater relative error measurement	Session 3	1.06 ± 0.07	3.77	1.00 ± 0	0.00
	Session 4	1.02 ± 0.06		1.00 ±0	

## Data Availability

The data presented in this study are available on request at the discretion of the corresponding author. The data are not publicly available due to privacy reasons.

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
