# Peer review of "SANE (Easy Gait Analysis System): Towards an AI-Assisted Automatic Gait-Analysis"

_ijerph, 2022, doi:10.3390/ijerph191610032_

Round 1
Reviewer 1 Report
Authors presented a markerless approach using AI for motion analysis during gait. Results are validated through two separate motion tracking systems. The goal of the project was to minimize human gait error. This article provides good materials in the field of AI in human biomechanics; relevant literatures and findings have been discussed; few minor comments are provided below to improve the study:
1. Introduction contains lots of relevant information on various topics such as use of peripherals: EMG, pressure plates, inertial sensors etc. It is recommended that authors categorize the content within the introduction section. Breaking it down into few subsections will help content flow better.
2. Description of the system: tools could be easily explained with descriptive mages.
3. Is there a source for equation 1 or justification to support?
4. The approach as shown equation (2), it focuses on position-based analysis, information why or why not orientation considered would be interesting to generalize the complete pose.
5. Adding the specific signal type and name on the arrows of figure 1, would be good. Specially the SANE Directory has multiple way communications. What are the specific algorithms in their would clearly address the AI need and its effectiveness?
6. In results section, instead of highlighting the data in tables, you may use bold font.
7. The axes values for most graphs do not have units and should be added.
8. In the conclusion section, authors state that one of the future objectives will be expanding pattern recognition capabilities of the system so that it can adapt to tracking other regions of the body. Given that, authors should also consider algorithms in which natural movement of human body can be differentiated from unorthodox movement of robots.
Reviewer 2 Report
The authors present a comprehensive description of their work, as a result, the paper is lengthy. In my opinion, the detailed information about the used software framework, and application operation could be skipped, if the authors do not provide their software in a software repository for the readers to reuse. I encourage authors to publish software code publicly, if they are not proprietary .
The presented results are focused on measurment repeatability and they take repeatability of the results as a measure of quality. For two reasons, it is not adequate in this case. First, each trial is different as a diagnosed person might perform differently. Second, this approch does not detect biased errors. It would be useful to compare SANE to the ground truth results. Perhaps, it would be a correct to take BTS system (expensive and spophisticated) as a ground truth and assess the SANE performance based on the comparison.
Reviewer 3 Report
In this paper, the authors proposed a gait analysis system called SANE (easy gAit aNalysis systEm) based on AI-Assisted. This research attempts to achieve a low cost for real-time analysis. The manuscript was well-reviewed about the related studies. However, this paper's originality and quality need to be improved thoroughly. Some questions and comments are the following:
1. This paper seems to combine many technologies and AI libraries to create a framework. Could the authors emphasize what the technical originality is by comparing related research? Because the reviewer thinks there would be no originality when only combining a pipeline. In addition, software like Matlab and libraries in python like SciPy need not be written in the manuscript.
2. Because this journal's aims relate to public health, what are this study's contributions to public health? It seems no biomechanics meanings have been investigated.
3. What are the meanings of the Right part in Fig. 2 ? The reviewer couldn't see useful information here. The reviewer only can know the left is human body detection.
4. On the right of Fig. 3 (b), where is the origin for spine x, y, z, and there is no x-, y-, z-axis draw in the fig? This also needs to be mentioned and defined here. The reviewer can see the text description on line 263. Why do authors choose ankle distance? And what is the mathematical definition of ankle trend? What is the difference between ankle distance and ankle trend? Also, the L2 Norm of spine x y z was calculated in eq. 2. What is this aiming for the analysis? The energy used to perfume the motions? How can users use eq.2 to analyze a gait based on its biomechanics meanings?
5. No legends in fig 6, so readers can not understand what the green and red lines stand for. What is the x-, and y-axis means in Figs. 5-10? There is also no label here to describe them.
6. All figures, including the GUI of the proposed method, are difficult to see. Please change the font size to make the figures more clear. Also, such as figures 4 and figure 1 have too many blanks. The layout also needs to be adjusted.
7. What does the numbers in front of each word? For example, line 93, 98, 100, 131, line 257, 258, 261, 264.
8. Why only the spine and ankle are enough to evaluate the system? The reviewer thinks that arm movements are also important for analyzing gait movement.
9. For the "Ankle Distance" trend, how about some movements that do not make orthogonal to the camera? The Ankle Distance can also be calculated correctly?
10. In eq.1, the FPS is not a unit. Do the authors mean Hz? And the maximum Hz does not need to be written as an equation. Maybe the authors could only mention it in the text is enough. If some limitations of this method can be listed quantitatively, a table may be a good choice.
11. In eq. 3, No G0 definition in the text. Please ensure all mathematical notations in the equation have been defined somewhere.
12. The comparison results with other capture systems look good. However, How about Messing points since this system uses interpolation, such as in figures 5 and 6. Will the system work correctly when difficult motion is performed at different angles?
Round 2
Reviewer 3 Report
The quality of the presentation, especially the figures, could still improve. However, the authors have responded to my concerns. The reviewer hopes the authors can improve these issues in the next work.
Author Response
Thank you for the appreciation of the work. The paper has gone through a full review and it has been better improved thanks to reviewers and academic editor editor suggestions.